# Ultra-High-Resolution Electrocardiography Enables Earlier Detection of Transmural and Subendocardial Myocardial Ischemia Compared to Conventional Electrocardiography

**DOI:** 10.3390/diagnostics13172795

**Published:** 2023-08-29

**Authors:** Kirill V. Zaichenko, Anna A. Kordyukova, Dmitry L. Sonin, Michael M. Galagudza

**Affiliations:** 1Laboratory of Radio- and Optoelectronic Devices for Early Diagnostics of Living Systems Pathologies, The Institute for Analytical Instrumentation, Russian Academy of Sciences, 31-33A Ivana Chernykh Street, 198095 Saint Petersburg, Russia; kvz24@mail.ru (K.V.Z.); sonin_dl@almazovcentre.ru (D.L.S.); 2Department of Microcirculation and Myocardial Metabolism, Institute of Experimental Medicine, Almazov National Medical Research Centre, 15B Parkhomenko Street, 194021 Saint Petersburg, Russia

**Keywords:** heart, ischemia, reperfusion, electrocardiography, ultra-high-resolution electrocardiography, power spectral density, ischemic heart disease, subclinical atherosclerosis, sensitivity, risk stratification

## Abstract

The sensitivity of exercise ECG is marginally sufficient for the detection of mild reduction of coronary blood flow in patients with early coronary atherosclerosis. Here, we describe the application of a new technique of ECG registration/analysis—ultra-high-resolution ECG (UHR ECG)—for early detection of myocardial ischemia (MIS). The utility of UHR ECG vs. conventional ECG (C ECG) was tested in anesthetized rats and pigs. Transmural MIS was induced in rats by the ligation of the left coronary artery (CA). In pigs, subendocardial ischemia of a variable extent was produced by stepwise inflation of a balloon within the right CA, causing a 25–100% reduction of its lumen. In rats, a reduction in power spectral density (PSD) in the high-frequency (HF) channel of UHR ECG was registered at 60 s after ischemia (power 0.81 ± 0.14 vs. 1.25 ± 0.12 mW at baseline, *p* < 0.01). This was not accompanied by any ST segment elevation on C ECG. In pigs, PSD in the HF channel of UHR ECG was significantly decreased at a 25% reduction of CA lumen, while the ST segment on C ECG remained unchanged. In conclusion, UHR ECG enabled earlier detection of transmural MIS compared to C ECG. PSD in the HF channel of UHR ECG demonstrated greater sensitivity in the settings of subendocardial ischemia.

## 1. Introduction

Along with cancer, ischemic heart disease (IHD) is one of the leading causes of mortality and morbidity worldwide. Despite significant improvement in the diagnosis, treatment, and prevention of IHD, its global prevalence has gradually increased over the last 30 years [1]. For instance, in total, there were 126.5 million patients with IHD in 2017; in the same year, IHD accounted for more than 8.9 million of deaths [2]. In the overwhelming majority of cases, IHD is caused by coronary atherosclerosis [3]. Although the rate of atherosclerotic plaque growth could vary depending on several non-modifiable and modifiable risk factors, in general it is quite slow. The extent of atherosclerosis and the volume of plaque greatly determine the clinical manifestations of IHD. The very early pathological manifestations of coronary atherosclerosis usually do not cause any symptoms and are commonly referred to as subclinical atherosclerosis (SCA) [4]. With the progression of coronary artery stenosis, the affected person may start to experience the classical symptom—acute chest pain—which usually occurs during physical exercise and/or emotional stress. The appearance of anginal pain is currently the main factor stimulating the patient to seek medical advice, resulting in clinical and diagnostic evaluation, which includes electrocardiography (ECG) at rest and during exercise, as well as echocardiography. When positive, the results of these diagnostic procedures confirm the presence of IHD. Conventional ECG (C ECG), both at rest and during physical exercise, is also included in the guidelines for large-scale health screening programs. At the same time, it should be emphasized that the existing ECG criteria for subendocardial ischemia including horizontal (or downslope) ST segment depression, flattening (or inversion) of the T wave, as well as the dynamics of these changes, are characterized by relatively low diagnostic sensitivity (~68%) and specificity (77%), thereby requiring revision [5]. An analogous situation is observed in acute coronary syndrome, where diagnostic sensitivity of C ECG is equal to 56% [6]. This evidence indicates that early signs of myocardial ischemia might be overlooked during routine ECG screening in a significant proportion of patients with SCA. As a result, this cohort of patients is not considered as having an increased risk of IHD in the future. The lack of medical advice regarding lifestyle modification and preventive drug therapy, as well as the absence of regular follow-ups would result in progression of disease, thereby moving it from a potentially reversible stage to irreversible one. Solid evidence indicates that the patients with SCA may benefit from active preventive measures (e.g., smoking cessation, physical exercise, modification of diet, and use of hypolipidemic therapy) due to slowing the process of coronary artery stenosis, thereby halting the progression of cardiovascular disease continuum to the next stage, which is overt IHD with stable/unstable angina [7]. In order to solve the problem, various approaches are currently being investigated. In particular, early signs of compromised myocardial perfusion in asymptomatic patients could be visualized using cardiac perfusion and metabolic scanning in positron emission tomography (PET) imaging [8]. It is evident, however, that the cost of PET imaging represents a major obstacle for its wide adoption as a screening tool for early IHD. One more technique is based on the quantification of coronary artery calcium using non-contrast cardiac computed tomography [9], but it is also characterized by low cost-effectiveness.

On the basis of the above evidence, it is clear that ECG is the only cost-effective method for predicting IHD. Apart from the generally used C ECG, there are at least three other types of ECG, which are aimed at the extraction of additional diagnostic information. In particular, high-resolution ECG is focused on low-voltage (~1 µV) late potentials which follow the QRS complex [10]. Late potentials reflect mild disorders of intraventricular conduction, serving as an important tool for the prediction of arrhythmic events in special cohorts [11]. One more type of ECG is high-frequency QRS ECG, which has been proven to be instructive in patients with IHD due to an analysis of the high-frequency (≤300 Hz) component of cardiac electrical signal [12]. In this study, we describe the new equipment for the registration of cardiac electrical activity, as well as the original approach to signal processing, which provides additional important diagnostic information in terms of early detection of ischemia-induced electrophysiological changes in the heart. This technique is termed ultra-high-resolution ECG (UHR ECG) [13,14]. UHR ECG is defined as the type of ECG which aims for the registration and analysis of cardiac electrical signals in the range of frequencies 0.05–2000 Hz with amplitude coverage down to 10 nV. In comparison to UHR ECG, high-frequency QRS ECG targets relatively narrow window of frequencies between 100 and 300 Hz while UHR ECG is operative in the extended range of frequencies. The method of UHR ECG is based in the original algorithm of mathematical processing of cardiac electrical signals registered in the extended amplitude and frequency ranges. Special algorithms of signal processing enable the calculation of different parameters including power spectral density (PSD) in the high-frequency (HF) channel. We hypothesized that UHR ECG might aid in the detection of subendocardial ischemia with greater sensitivity and specificity than conventional ECGs (C ECGs). We predict that the introduction of UHR ECG as a part of the program of medical surveillance would contribute to earlier detection of SCA in asymptomatic persons undergoing health screening. This might lead to a significant improvement of cardiovascular risk stratification, better primary prevention of IHD, and, potentially, a reduced number of advanced cases associated with increased mortality. In order to validate this hypothesis and compare the diagnostic potential of UHR ECG versus C ECG, two series of experiments have been performed on anesthetized rats and pigs. In rats, the time until the appearance of typical ECG changes after induction of zero-flow transmural myocardial ischemia has been measured. In pigs, subendocardial ischemia of a variable extent was produced by stepwise inflation of a balloon placed in the right coronary artery. In this series, we registered the extent of coronary flow reduction sufficient for the appearance of changes in main variables on ECG UHR ECG and C ECG. Thus, the results obtained confirmed the hypothesis that UHR ECG enables earlier recognition of both transmural and subendocardial ischemia.

## 2. Materials and Methods

### 2.1. Ethics Statement

All procedures were performed in accordance with the Guide for the Care and Use of Laboratory Animals (NIH publication No. 85–23, revised 1996) and the European Convention for the Protection of Vertebrate Animals used for Experimental and other Scientific Purposes. The Institutional Animal Care and Use Committee at the Almazov National Medical Research Centre approved the study protocol (Protocol Number PZ_22_3_SoninDL_V3, 23 March 2023). All efforts were made to protect the animals and minimize their suffering during the study. The experiments complied with the ARRIVE guidelines (http://www.nc3rs.org/ARRIVE, accessed on 26 July 2023).

### 2.2. Animals

The experiments were performed on anesthetized rats and pigs. Male Wistar rats (age, 12–14 weeks; weight, 320–350 g) of SPF grade were obtained for the study from the SPF breeding facility (Pushchino, Moscow, Russia). The animals were maintained in individually ventilated cages under a 12/12 h light/dark cycle and were given free access to food and water. Subendocardial ischemia was induced in domestic female Landrace pigs weighting 50–60 kg. All pigs were purchased from the same breeding farm. Pigs were individually housed in the pens with a minimum of 3.5 × 5.5 ft floor space. Pen walls were completely solid; all front gates were made of stainless steel vertical bars. Drinking water and commercially available diet were provided ad libitum. Ambient temperature and humidity were maintained at 18–28 °C and 25.0–85.0%, respectively.

### 2.3. The Model of Acute Transmural Myocardial Ischemia in the Rat

The animals were placed in the acrylic plastic chamber for anesthesia induction with 3.0% isoflurane (Forane; Abbott Laboratories Ltd., Queenborough, Kent, UK). After induction, the oxygen-enriched (~30%) gas was delivered through a facial mask at 1.2–1.5% using low-flow gas apparatus (SomnoSuite; Kent Scientific, Torrington, CT, USA). The animals were fixed in the supine position on a feedback-controlled heating pad (TCAT-2LV; Physitemp Instruments Inc., Clifton, NJ, USA). Body temperature was maintained at 37.0 ± 0.5 °C. Through midline cervical incision, the common carotid artery was accessed, followed by its cannulation with polyethylene catheter. The heparinized catheter was connected to a commercial data acquisition system for continuous monitoring of mean arterial pressure (MAP) and heart rate (HR) (PhysExp Gold; Cardioprotect Ltd., St.-Petersburg, Russia). In addition, the right jugular vein was cannulated for drug injection. After that, the animals were switched to mechanical ventilation via tracheotomy (SAR–830P; CWE, Inc., Ardmore, PA, USA). After skin incision and muscle retraction, the thorax was opened via lateral incision in the fourth intercostal space. The pericardium was removed, followed by identification of the left coronary artery (LCA). A 6–0 polypropylene ligature was placed around the proximal segment of the LCA, and the occluder was made by passing the ends of the thread through a 10.0 cm polyethylene tube [15]. The electrodes of C ECG (Cardiotechnica-ECG-8; Incart Ltd., St.-Petersburg, Russia) and UHR ECG were implanted subcutaneously according to a standard scheme. The registration of both C ECG and UHR ECG was performed continuously throughout stabilization and a 10 min period of transmural zero-flow ischemia elicited by LCA occlusion in the standard leads I-III. PSD in the HF channel of the UHR ECG was calculated on the bases of the signal obtained from lead II because this particular lead is more sensitive in terms of ischemia of the anterolateral wall of the left ventricle (LV). After completion of 10 min ischemia, the bilateral thoracotomy was completed, followed by intravenous administration of 0.5 mL of 1% Evans blue and heart excision. After rinsing in 0.9% NaCl solution, the heart was cut into 5 transverse slices. The slices were photographed using a digital camera for visualization and quantification of the anatomical area at risk (AAR) (ImageJ 1.34s; National Institutes of Health, Bethesda, MD, USA), which was defined as Evans blue-negative area [16]. The hearts with AAR of less than 15% were excluded from the analysis.

### 2.4. The Model of Gradual Subendocardial Ischemia in Pigs

The animals (*n* = 3) were deprived of food for 24 h prior to surgery. Anesthesia was induced with tiletamine-zolazepam at a dose of 20 mg/kg, i.m. (Zoletil 100; Virbac, France), xylazine at a dose of 3 mg/kg, i.m. (Xyla; Interchemie, Netherlands), and atropine at a dose of 0.1 mg/kg, s.c. (Atropin, Moscow Endocrine Factory, Moscow, Russia). The animals were intubated and mechanically ventilated at a tidal volume of 25–30 mL/kg/min (respiratory rate-20–25/min, 65% oxygen). After intubation, the isoflurane was added to the gas mixture at 2%. This was followed by cannulation of the superficial ear vein and implantation of subcutaneous electrodes for continuous monitoring of C ECG and UHR ECG in the standard leads I-III. The calculation of PSD and assessment of the ST segment deviations were performed in the lead III, which best reflects ischemic changes in the portion of the right ventricle belonging to the posterior heart surface. Core body temperature was monitored using the rectal probe. Mean arterial pressure and heart rate were measured via the catheterization of the aorta. The animals received heparin (300 IU/kg) under the control of activated clotting time. Subendocardial ischemia was induced by angiography-guided positioning of the balloon catheter in the lumen of right coronary artery (RCA) [17]. Briefly, the guide catheter was introduced through the femoral artery and placed into the lumen of the descending branch of the RCA, followed by positioning of the 6FR coronary balloon dilatation catheter (OTW; Terumo, Tokyo, Japan). After registration of baseline parameters, the balloon was inflated in a stepwise manner in order to produce 25, 50, 75, and 100% reduction in RCA lumen followed by complete reperfusion via balloon deflation. Each step lasted for 5 min.

### 2.5. Registration of UHR ECG and Signal Processing

In this study, we used the original method of UHR ECG, which might be superior to conventional ECG in certain aspects [18,19]. UHR ECG ensures registration and analysis of all meaningful changes of the electrical cardiac potentials using standard ECG electrodes. This is achieved by inclusion of maximally extended frequency and magnitude ranges [20]. In particular, UHR ECG is aiming to analyze electrical activity of the heart in the frequency range with the upper limit of 2000 Hz (Figure 1a). At the same time, the lower input voltage border in UHR ECG is extended down to 10 nV, which is only possible by using modern equipment with enhanced sensitivity of the recording device. Thus, UHR ECG holds the advantage of accurately detecting all conventional characteristics of ECG while providing additional indispensable information primarily derived from low-voltage and high-frequency (HF) components of cardiac electrical activity (Figure 1a). The method of UHR ECG incorporates detailed spectral and correlation analysis of cardiac potentials [21,22]. After amplification, cardiac electrical signal from the electrodes is subdivided into low-frequency (LF) and HF components according to predefined borderline values of low (f_l_) and upper (f_u_) frequency in the corresponding LF and HF channels (Figure 1b). After additional amplification, the signals from both channels are transmitted to the analogous-to-digital converter and personal computer. The system is equipped with computer-based control of the characteristics of band filters for HF signal as well as of the amplification process through the digital-analog converter. Figure 1c illustrates the frequency-based characteristics of the module for two-channel analogous processing of UHR ECG while Figure 1d shows oscillograms of the signals registered in different locations of the module.

Special mathematical processing of the signal obtained from the HF channel includes Fourier transform from the signal function, as well as the calculation of power spectral density (PSD) S(f) of the signal in the HF channel of UHR ECG. This is the function describing the distribution of power of the HF signal depending on the frequency, i.e., the power falling within the defined frequency (f) interval. In order to quantitatively characterize the changes of PSD in the HF channel, we use the physical value—power P—of the signal in the entire band of frequencies of the HF channel 100–1000 Hz.

### 2.6. Statistical Analysis

Statistical analysis was performed using SPSS 12.0 (IBM Corporation, Armonk, NY, USA). The Kruskal–Wallis test was used to determine differences in hemodynamic parameters and ECG parameters between various timepoints. This was followed by pairwise interpoint comparisons that were performed using the nonparametric Mann–Whitney U test. *p* values < 0.05 were considered statistically significant.

## 3. Results

### 3.1. Verification of Transmural and Subendocardial Ischemia

In rats (*n* = 24), the presence of regional myocardial ischemia after LCA occlusion was verified by the changed color of the epicardium in the ischemia area, ST segment elevation (both in the LF channel of UHR ECG and C ECG), and the occurrence of early ischemic tachyarrhythmias. The parameters of ischemic tachyarrhythmias are presented in Table 1. Mechanical defibrillation was performed in all animals developing episodes of ventricular fibrillation (VF). In four animals, defibrillation was ineffective. These animals were not included in the analysis of AAR. Representative examples of ischemic tachyarrhythmias, that is, ventricular tachycardia and ventricular fibrillation, are shown in Figure 2.

The presence of reproducible transmural ischemia was also verified by the estimation of AAR. The AAR size was 42 ± 7% (Figure 3). In two experiments, the AAR size was smaller than 15%. These experiments were excluded from the final analysis.

The onset of regional myocardial ischemia in pigs was verified using such C ECG changes as ST segment shift from the isoelectric line and/or appearance of a negative T wave. The correct position of the occluding intravascular balloon in the RCA and the extent of its inflation were controlled using angiographic imaging.

### 3.2. Physiological Monitoring

For better standardization of ECG data, regular measurement of basic physiological parameters was performed in all animals. In rats, the values of MAP, HR, and core body temperature were not different across the course of the experiment (Table 2). Moreover, all parameters were within the normal range. The same was true for experiments in swine. Prior to ischemia, the values of systolic blood pressure, diastolic blood pressure, HR, and body temperature were 87 ± 6 mm Hg, 47 ± 9 mm Hg, 99 ± 13 beats per min, and 37.1 ± 0.8 °C, respectively. After ischemia, the hemodynamics remained stable.

### 3.3. ECG Detection of Transmural Ischemia in Rats

Transmural ischemia was detected using C ECG and UHR ECG in rat experiments (Figure 4a). The morphologies of C ECG and UHR ECG in the LF channel were generally similar. Significant elevation of the ST segment was observed starting from the third minute of myocardial ischemia (Figure 4c,e,g), while a significant reduction in PSD in the HF channel of UHR ECG was registered at the first minute of ischemia (*p* < 0.01, Figure 4b,d,f). The baseline value of signal power P in the HF channel was 1.25 ± 0.12 mW (Figure 4b,d). At 1 min of ischemia, it was decreased to 0.81 ± 0.14 mW (*p* < 0.01 vs. baseline, Figure 4b,f). At 3 min of ischemia, it was further significantly decreased to 0.37 ± 0.11 mW (*p* < 0.01 vs. 1 min, Figure 4b,h).

### 3.4. ECG Detection of Subendocardial and Transmural Ischemia in Pigs

Subendocardial and transmural ischemia were induced in pigs using an endovascular balloon placed in the lumen of the RCA. The balloon was inflated in a stepwise manner in order to produce 25, 50, 75, and 100% reduction of the coronary flow with subsequent reperfusion via complete deflation. In this model, coronary flow reduction by 25–75% corresponded to subendocardial ischemia, and by 100% it corresponded to transmural ischemia. The extent of downslope depression of the ST segment in millimeters in the lead II was estimated on C ECG and the LF channel of UHR ECG at baseline and at all stages of coronary flow reduction on 5th minute. At the same timepoints, we also registered the PSD, which was expressed as the signal power P (mW) in the HF channel of UHR ECG (Table 3). It should be noted that a 25% reduction of flow, which corresponds to the minimal extent of subendocardial ischemia, was not associated with any deviation of the ST segment from the isoelectric line (Table 3, Figure 5e). It is noteworthy that PSD at 25% reduction was significantly lower in comparison to that at baseline (Figure 5b,f). With an increased extent of coronary flow reduction (50–75%), a clear-cut downslope depression of the ST segment has become evident on C ECG and in the LF channel of UHR ECG (Table 3, Figure 5g,i). This was paralleled by proportional decrease in the PSD in the HF channel of UHR ECG (Table 3, Figure 5b,h,j). In 100% flow reduction, which corresponds to the transmural ischemia, there were no ST segment elevations observed, which might have been accounted for by the insufficient ischemia duration and also by the protective effect of ischemic preconditioning induced by stepwise reduction of blood flow from 100 to 0% with the step of 25%. It is noteworthy that at reperfusion we observed some extent of PSD normalization (signal power P in the HF channel of UHR ECG was equal to 0.37 ± 0.04 mW, *p* < 0.05 vs. baseline and 100% reduction, Table 3, Figure 5b,n). At reperfusion, the ST segment tended to become less depressed compared to 100% flow reduction (Table 3, Figure 5m).

## 4. Discussion

The major finding of the present work is that new method of UHR ECG has made it possible to detect the presence of myocardial ischemia earlier than C ECG. Two different models of ischemia have been used to evaluate the diagnostic potential of UHR ECG. Transmural ischemia occurs in critical stenosis or in complete occlusion of infarct-related coronary artery in patients with acute coronary syndrome [23]. In the experimental settings, transmural ischemia is reliably induced by means of coronary artery occlusion in laboratory rodents. ST segment elevation in the leads corresponding to infarct location is known to be a classical ECG sign of transmural ischemia in the acute period. Reciprocal changes of the ST segment are registered in the leads from the opposite to the infarct wall of the LV. It is generally accepted that ST segment elevation is explained by permanent partial depolarization of the ischemic portion of the myocardium, which takes place also during the resting phase, at which all other parts of the heart are repolarized [24]. In this situation, the net electrical vector is directed from the active (positive) electrode, resulting in the downward shift of all elements of ECG except the ST segment. Zero voltage is registered only during general myocardial depolarization which corresponds to the ST segment; this zero voltage is interpreted as ST segment elevation [25]. Our rat experiments also demonstrated significant ST segment elevation (>3 mm), which was evident on C ECG and also in the LF channel of UHR ECG at 180 s after ischemia onset. However, we registered significantly reduced PSD in the HF channel of UHR ECG already at 60 s of ischemia. Importantly, at that moment, the morphologies of the QRS wave and the ST segment had no differences from those in baseline ECG. Collectively, these data indicate that PSD in the HF channel of UHR ECG is more sensitive to mild disorders of cardiac ionic homeostasis occurring during the first minute of ischemia.

A possibility exists that PSD in the HF channel of UHR ECG is becoming decreased in response to the smaller extent of ischemic depolarization linked to enhanced inward Na^+^ current and associated K^+^ leakage secondary to Na^+^/K^+^-ATPase dysfunction, at which ST segment elevation has not yet started. The findings on transmural ischemia detection using UHR ECG are of interest in terms of fundamental mechanisms. Additional studies will be required to define the diagnostic value of UHR ECG in patients with acute coronary syndrome. At present, acute coronary syndrome is diagnosed on the basis of clinical, electrocardiographic, and biochemical criteria in the period when all typical changes on C ECG are already evident [26].

Compared to transmural ischemia, the application of UHR ECG for detection of mild to moderate reduction of coronary blood flow in subendocardial ischemia may be more practically relevant. In the clinical realm, subendocardial ischemia may develop in persons with a minimal degree of coronary blood flow reduction, usually in association with markedly increased myocardial oxygen consumption secondary to intense physical exercise or inotropic challenge [27]. The mechanistic model of subendocardial ischemia development is based on the combined effect of reduced perfusion pressure in the affected vascular bed and systolic compression of penetrating arteries connecting large subepicardial arteries to the subendocardial vascular plexus. Classic ECG signs of subendocardial ischemia comprise either horizontal or downslope ST segment depression, flattening of the T wave and/or its inversion, as well as the evolution of these changes with time [28]. However, these signs are characterized by relatively low diagnostic sensitivity (~68%) and specificity (77%); in addition, topical diagnostics of ischemia using such criteria are unreliable [5]. In this regard, any effort aiming to increase the ability of ECG to discriminate between normal and reduced endocardial blood flow deserves careful consideration. In this study, subendocardial ischemia was produced in swine using the controlled reduction of the cross-sectional area of RCA. This model seems to be optimal for such experimental design since it closely resembles the clinical scenario of subendocardial ischemia [29]. Upon stepwise reduction of coronary blood flow in the RCA, we observed typical changes on C ECG, including downslope depression of the ST segment. It is noteworthy that the extent of ST segment depression correlated with the degree of flow restriction. Particularly, reduction of flow by 25% was associated with significant decrease in PSD in the HF channel of UHR ECG, while it had not yet caused measurable ST segment depression. This observation points to the conclusion that UHR ECG enables the detection of early, reversible electrophysiological alterations in the heart occurring at minimally reduced blood flow. Since UHR ECG is relatively inexpensive, it might become an indispensable tool for screening and identification of persons with SCA, necessitating implementation of populational and individual prevention strategies in this cohort. It should be noted that numerous studies exploring the utility of C ECG for prediction of coronary events in healthy persons and in patients with mild IHD have been performed previously. The results of these studies are somewhat contradictory. Initial studies dated back to 1990s generally showed that ST segment depression on exercise testing in apparently healthy individuals could not predict the development of angina, myocardial infarction, or cardiovascular death in future [30]. Subsequently, it has been demonstrated that ST segment depression ≥ 1.00 mm on exercise testing in asymptomatic persons was associated with a significantly increased risk of coronary events in the following 7.9 years [31]. This was later confirmed. For instance, the presence of even minimal ST segment depression in healthy individuals was accompanied by an increased risk of all-cause mortality in a prospective study lasting for 13.7 and 13.9 years for males and females, respectively [32]. Nonetheless, the use of novel calculated parameter in the form of PSD in the HF channel, as it was demonstrated for the first time in the present work, or some other UHR ECG-derived markers of ischemia, could potentially provide a better IHD risk stratification in asymptomatic middle-aged individuals than the dynamics of the ST segment on C ECG.

This preliminary study has several limitations. First, the number of experiments performed on pigs is quite low. Nevertheless, we feel that the first findings provide important clues for future large-scale experimental validation studies. There are other remarkable perspectives on the studies on diagnostic utility of UHR ECG. In particular, we plan to set the continuous online calculation of PSD changes in the HF channel for more precise analysis of its dynamic changes, which might be critical for early detection of ischemia. In addition, the diagnostic sensitivity and specificity of UHR ECG in terms of ischemia recognition should be estimated. The optimization of technical, hardware, and algorithmic components of the method would contribute to better diagnostic sensitivity and robustness of UHR ECG. Finally, after corroboration of the results of the present study, we plan to elaborate the UHR ECG-based algorithm of early detection of SCA, and to perform a clinical study of the efficacy of this algorithm in healthy volunteers aged 45–50 years. The participants will be subjected to exercise testing in combination with UHR ECG registration, followed by cardiorespiratory exercise test with the measurement of lactate in the plasma.

## 5. Conclusions

In this work, we describe for the first time the general technical principles of UHR ECG, as well as the approaches to program-based analysis of the UHR ECG signals in the HF channel. It was demonstrated that UHR ECG, coupled with mathematical analysis of recorded signals, enables earlier detection of acute transmural ischemia and increases the sensitivity of subendocardial ischemia detection. PSD in the HF channel of UHR ECG has proved to be the most instructive calculated parameter for early recognition of ischemia. Earlier detection of subendocardial ischemia in patients with SCA opens the perspective of using UHR ECG for cost-effective diagnostics of IHD at potentially preventable stages.

## Figures and Tables

**Figure 1 diagnostics-13-02795-f001:**
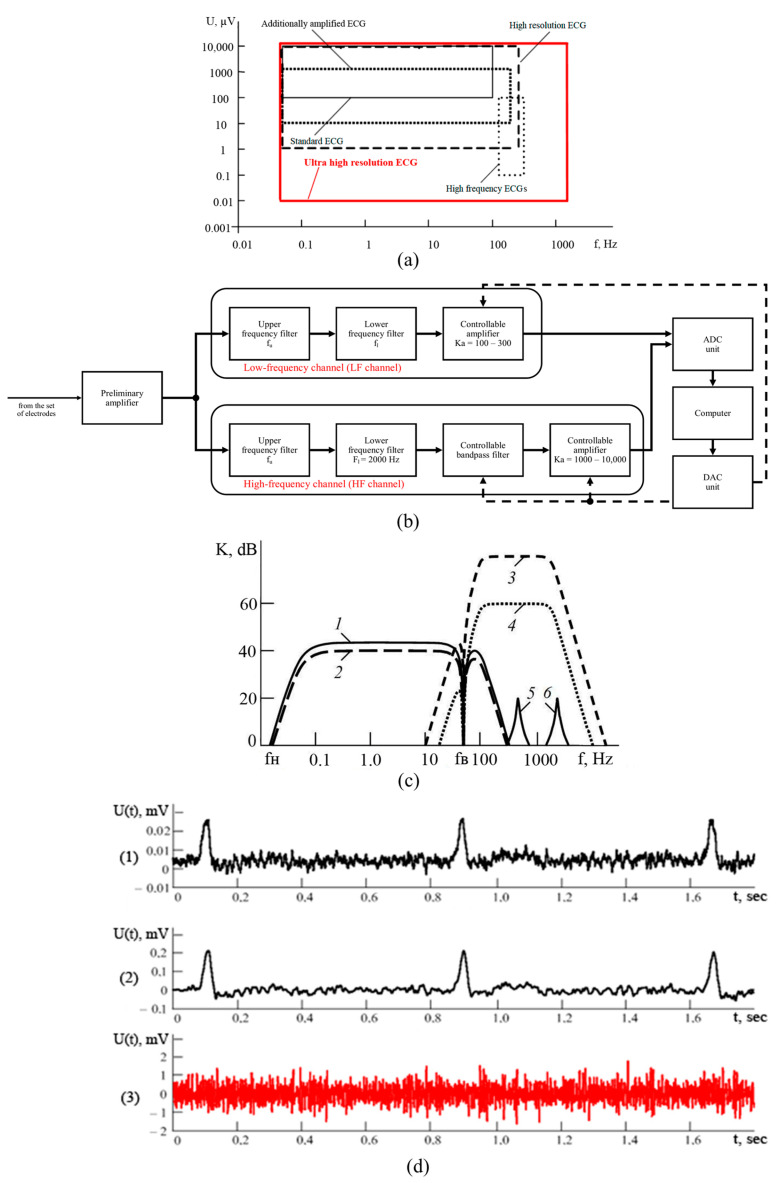
New technique of UHR ECG. (**a**) Comparison of magnitude (U, μV) and frequency (f) ranges used for the analysis in different types of ECG; (**b**) the scheme of two-channel signal processing system utilized in UHR ECG; (**c**) the graph demonstrating the values of transmission coefficient (K) expressed in decibels (dB) depending on the frequency of different channels of the analogous ECG processing device according to UHR ECG method; examples of frequency characteristics of different channels include low-frequency (1, 2), high-frequency (3, 4), and controlled band filters (5, 6); (**d**) oscillograms showing time-dependent changes in voltage (U(t)) of the signals in the module of analogous signal processing according to UHR ECG, i.e., the signal at the input to the module (plot 1), the signal at the output of the LF channel (plot 2), and the signal at the output of the HF channel (plot 3, in red).

**Figure 2 diagnostics-13-02795-f002:**
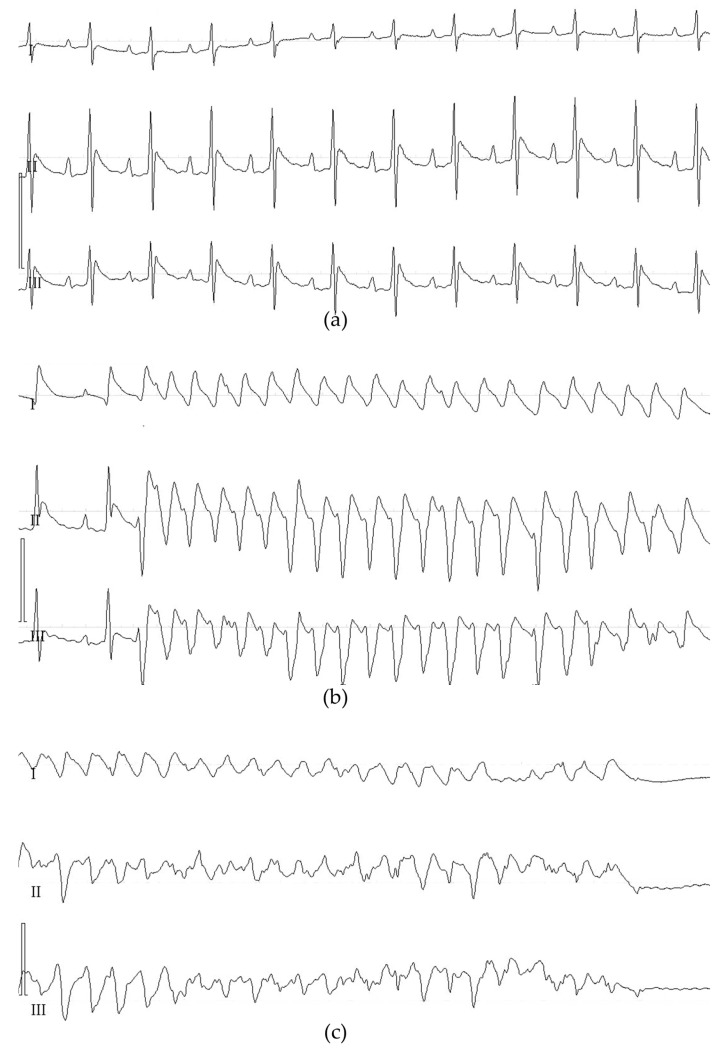
Representative C ECG recordings in the standard leads (I-III) showing ischemic ventricular tachyarrhythmias in rats with LCA occlusion. ECG was registered in the standard leads at a speed of 100.0 mm/s. (**a**) ECG at baseline; (**b**) an episode of ventricular tachycardia; and (**c**) ventricular fibrillation with asystole.

**Figure 3 diagnostics-13-02795-f003:**
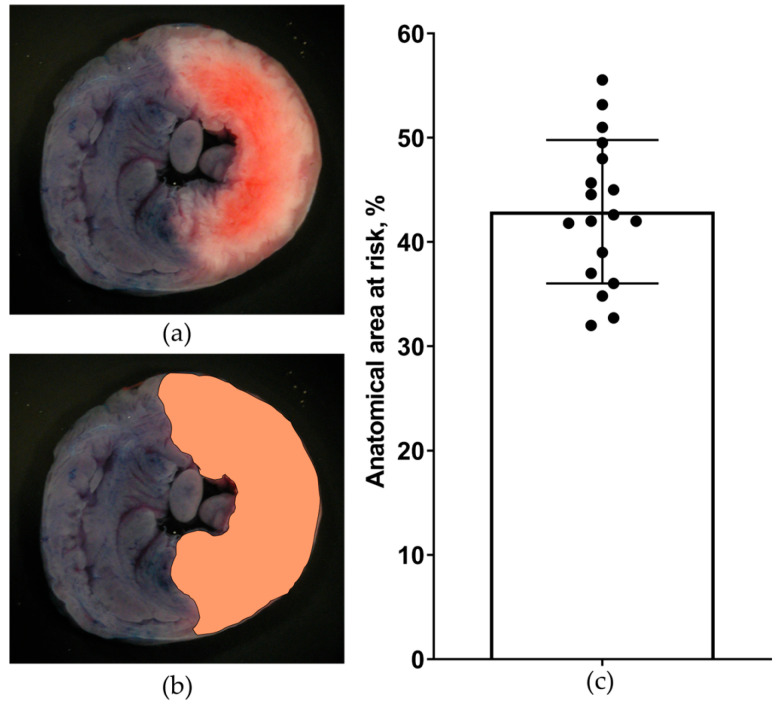
Anatomical area at risk detection and quantification in rats subjected to LCA occlusion (*n* = 18). (**a**) Representative transverse cardiac slice taken from Evans blue-stained heart showing the anatomical area at risk (Evans blue-negative tissue); (**b**) the same slice after software-assisted delineation of the anatomical area at risk; and (**c**) numerical data on the size of the anatomical area at risk expressed as a percentage of the entire surface of the slices.

**Figure 4 diagnostics-13-02795-f004:**
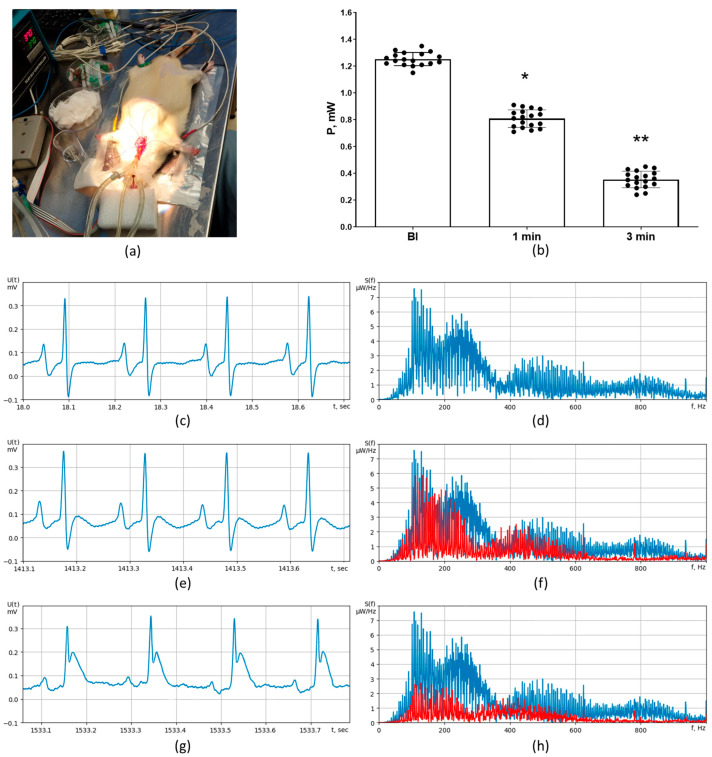
Detection of acute transmural ischemia in rats (*n* = 18) using C ECG (standard lead II) versus UHR ECG. (**a**) General view of the experimental setup; (**b**) numerical values of signal power P (mW) in the HF channel of UHR ECG at baseline and at different timepoints of ischemia; (**c**) representative C ECG recording at baseline; (**d**) representative pattern of power spectral density S(f) at baseline; (**e**) representative C ECG recording at 1 min after coronary occlusion; (**f**) representative pattern of power spectral density S(f) at 1 min after coronary occlusion (in red; baseline level is marked in blue); (**g**) representative C ECG recording at 3 min after coronary occlusion showing the ST segment elevation; (**h**) representative pattern of power spectral density S(f) at 3 min after coronary occlusion (in red; baseline level is marked in blue). * indicates *p* < 0.01 when data are compared to that for the baseline (BI); ** indicates *p* < 0.01 when data are compared to that for 1 min.

**Figure 5 diagnostics-13-02795-f005:**
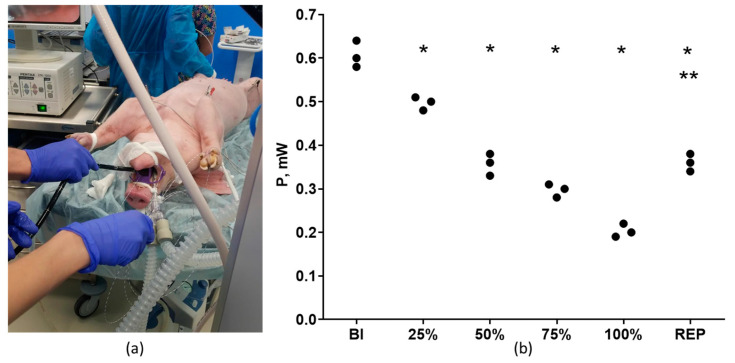
Detection of subendocardial and transmural ischemia in pigs (*n* = 3) using C ECG (standard lead III) versus UHR ECG. (**a**) General view of the experimental setup; (**b**) numerical values of signal power P (mW) in the HF channel of UHR ECG at baseline (BI) and at different grades of coronary flow reduction, as well as at reperfusion (REP); (**c**) representative C ECG recording at baseline; (**d**) representative pattern of power spectral density S(f) at baseline; (**e**) representative C ECG recording at 25% flow reduction; (**f**) representative pattern of power spectral density S(f) at 25% flow reduction (in red; baseline level is marked in blue); (**g**) representative C ECG recording at 50% flow reduction showing downslope ST segment depression; (**h**) representative pattern of power spectral density S(f) at 50% flow reduction; (**i**) representative C ECG recording at 75% flow reduction; (**j**) representative pattern of power spectral density S(f) at 75% flow reduction; (**k**) representative pattern of power spectral density S(f) at 100% flow reduction; (**l**) representative pattern of power spectral density S(f) at 100% flow reduction; (**m**) representative C ECG recording at reperfusion showing partial resolution of ST segment depression; (**n**) representative pattern of power spectral density S(f) at reperfusion (in orange; baseline level is marked in blue). * indicates *p* < 0.01 when data are compared to that for the baseline (BI); ** indicates *p* < 0.01 when data are compared to that for the 100% flow reduction.

**Table 1 diagnostics-13-02795-t001:** Characteristics of ischemic ventricular tachyarrhythmias (VTA) in rats subjected to LCA occlusion resulting in transmural myocardial ischemia. Data are mean ± standard deviation. VF—ventricular fibrillation.

VTA Characteristics	Value (*n* = 24)
Number of animals with VTA	22 (92%)
Number of VTA episodes per 1 animal	2.6 ± 0.8
Time to onset of the first episode of VTA, s	243 ± 45
Total duration of VTA, s	48 ± 22
Number of animals with persistent VF	4 (17%)

**Table 2 diagnostics-13-02795-t002:** Main physiological variables registered in rat experiments. Data are mean ± standard deviation. MAP—mean arterial pressure, HR—heart rate.

Physiological Parameters	Prior to Thoracotomy	5 min Prior to Ischemia	5 min Ischemia	10 min Ischemia
MAP, mm Hg	116 ± 12	118 ± 14	110 ± 9	112 ± 16
HR, beats/min	422 ± 25	406 ± 29	412 ± 32	401 ± 18
Body temperature, °C	37.2 ± 0.4	36.9 ± 0.6	37.0 ± 0.5	37.5 ± 0.7

**Table 3 diagnostics-13-02795-t003:** Maximal deviation of the ST segment from the isoelectric line and power spectral density in the HF channel of UHR ECG at different grades of coronary flow reduction through the RCA in pigs (*n* = 3). Data are mean ± standard deviation. * indicates *p* < 0.05 when data are compared to that for the baseline (BI); ** indicates *p* < 0.05 when data are compared to that for the 100% reduction.

Stage of the Experiment	Maximal ST Segment Deviation from the Isoelectric Line, mm	Signal Power P in the HF Channel of UHR ECG, mW
Baseline	0	0.61 ± 0.05
Reduction 25%	0.0 ± 0.05	0.52 ± 0.05 *
Reduction 50%	2.2 ± 0.21 *	0.36 ± 0.04 *
Reduction 75%	2.8 ± 0.34 *	0.31 ± 0.03 *
Reduction 100%	3.7 ± 0.42 *	0.20 ± 0.02 *
Complete reperfusion	1.9 ± 0.15 *^,^**	0.37 ± 0.04 *^,^**

## Data Availability

The data presented in this study are available on request from the corresponding author.

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
