# Peer review of "Ultra-High-Resolution Electrocardiography Enables Earlier Detection of Transmural and Subendocardial Myocardial Ischemia Compared to Conventional Electrocardiography"

_diagnostics, 2023, doi:10.3390/diagnostics13172795_

Round 1

Reviewer 1 Report

Dest Authors 

The manuscript was a valuable diagnostic study with clear title. Abstract was concise and precise and well structured. Introduction described comprehensively the problem. The methodology was correct in all domains from sampling size and sampling method to statistical data analysis and well explained that well designed study. Result was sound and clear. Discussion was appropriate and supportive for findings. Conclusion was consistent with discussion and results and finally references was written with a good manner.

Regards

Reviewer 2 Report

The authors have conducted a series of animal experiments to investigate the application of high-definition ECG and traditional ECG in the early diagnosis of myocardial ischemia. They have found that the high-frequency channel of ultra-high resolution ECG shows an advantage in early detection of myocardial ischemia. I have some minor comments:

1. There seems to be an issue with the units of the vertical coordinates in the three graphs presented in Figure 1. It is unclear whether the voltage of the high-frequency component is higher than that of the low-frequency component. Please verify and clarify this discrepancy.

2.It would be beneficial to provide a clear definition of ultra-high resolution electrocardiography (ECG) in the introduction. This will help readers unfamiliar with the technique to understand its principles and advantages.

3.It is important to clarify the distinction between ultra-high resolution ECG and high-frequency QRS ECG. Since high-frequency QRS ECG also holds significant value in the diagnosis of myocardial ischemia, it would be beneficial to conduct a comparative experiment to assess their respective effects. By comparing the performance of ultra-high resolution ECG and high-frequency QRS ECG, the authors can provide valuable insights into their relative merits and potential synergies.

Reviewer 3 Report

The paper is about showing usability of high frequency ECG in early detection of ischemic changes is coronary circulation.

Extremely well written paper. English is perfect. Very easy to read.

I just di not understand how many ECG leads were used and why. That affects the accuracy of the standard ECG. Also, is the power of the high frequencies taken for the whole measurement or for some segment. Does it make sense to calculate power only for ST-T interval for instance?
